# Sustainable Supplier's Equilibrium Discount Strategy under Random Demand

**Yingxiao Li and Jianheng Zhou ***

Glorious Sun School of Business and Management, Donghua University, Shanghai 200051, China;
liyingxiaoxy@163.com
* Correspondence: zjh001@dhu.edu.cn

**Abstract:** This paper examines a sustainable supplier's price discount strategy in a competitive environment as it considers building a two-level supply chain system consisting of two suppliers and a single retailer under the condition of uncertain demand, and investigates the impact of the suppliers' price discount strategy on the retailer's expected profits. We find that the sustainable supplier's expected profit increases as the price discount increases. When only the sustainable supplier offers a discount, the retailer's purchasing cost will increase with the degree of the discount; when both suppliers provide a discount, the sustainable supplier's expected profit decreases as the price discount increases, while the retailer's purchasing cost will decrease.

**Keywords:** price discount; supply chain; pricing; sustainability

## 1. Introduction

Sustainability has become a global issue, and various brands have established sustainable development goals and plans to reduce carbon emissions. For example, in September 2020, Walmart president and CEO Doug McMillon announced that Walmart will transform into a resource-renewable company and redouble its efforts to deal with the ever-increasing climate crisis, and established the goal of achieving zero emissions in the company's global business operations by 2040. Another example is that Apple is continuing to work hard to establish a 100% closed-loop supply chain, encouraging other companies to produce attentively, and striving to consolidate the company's environmental awareness. Furthermore, in order to increase the demand of sustainable products, some firms also provide price discounts to encourage retailers or consumers to purchase environment friendly products. Take Apple, for example, which provides Apple-certified refurbished products, which undergo a rigorous refurbishment process and are offered at special discounts of up to 15% compared to the price of a new product.

On the other hand, the rapid integration of traditional industries and "Internet +" promotes the reshaping of the industrial chain, supply chain, and value chain [1,2]. In order to compete for market share, many firms also offer price discounts for retailers to encourage them to purchase more products from them [3,4]. A good example is the 1688 platform (a wholesale platform under Alibaba)—a purchasing wholesale network in China—which offer stepped wholesale prices, which means the retailer will get a lower wholesale price when ordering more than a certain quantity.

To sum up, suppliers in the market may offer price discounts for two different purposes: sustainable suppliers provide price discounts to increase e sales of sustainable products, and competitive suppliers provide price discounts for competitive purposes.

In this paper, we are going to discuss the impact of quantity discounts. The competitive environment makes sustainable suppliers face a trade-off when deciding whether to provide price discounts: on the one hand, when a price discount is provided, retailers may purchase more from the sustainable supplier; on the other hand, other suppliers may also provide price discounts for competitive purposes when they are informed that the sustainable

supplier will provide a price discount: if they do, the competition of such price discounts will reduce each supplier's sales profits, but it can prevent the market being divided up; if not, the opposite is true [5,6].

Based on the above considerations, we develop a model in which a sustainable supplier considers providing price discounts to the retailer to promote the sustainable product sale. Meanwhile, another supplier which provides the same product may then be motivated to provide price discounts when it is informed that the sustainable supplier will provide price discounts. We explore the sustainable supplier's first-best quantity decision and price discount strategy, then we investigate the influence of suppliers' price discounts on supply chain members. We find that when only the sustainable supplier provides a price discount, their expected profit will increase due to the retailer ordering more from them, while the sustainable supplier's expected profit will decrease when both suppliers provide price discounts, because the game result of two suppliers offering price discounts is that both sides offer the same price discount. This leads to the sustainable supplier lowering the price on the one hand, while on the other hand the market demand does not increase, benefiting the retailer in this competitive game.

## 2. Literature Review

Supply chain competition has been extensively studied in the operations management literature [7–10]. Those studies mainly include two major directions: competitive means and supply chain structure. For example, Li and Wan (2017) [11] study the single channel supply chain competition problem, while Huang et al. (2018) [12] study the dual supply chain competition problem. The literature which studies the competition means include price discounts [13], investment strategy [14], inventory strategy [9], and so on. This paper studies how to formulate the optimal price discount strategy in the competitive environment.

In the literature on price discounts, an initial price discount is defined as a direct inducement that offers extra value or incentive for consumers with the primary objective of creating an immediate sale [15,16]. The price discount contract was originally used between retailers and consumers, and then introduced between suppliers and retailers [17]. The Advance Purchase Discount contract is used to encourage retailers to place their orders early [18,19]. Other research combines the price discount strategy with the consideration of supply chain members' special psychology, like retailers' unfair disgust [20], consumers' feelings [21,22], and other factors independent of members, like advertising strategy and so on [23,24].

Most of these studies combine the price discount problem with an exogenous strategy, or with another supply chain member's strategic behavior, and did not put the price discount problem in a competitive environment. In this paper, we are going to investigate the sustainable supplier's price discount strategy, while also considering that other suppliers may also provide price discounts due to the competitive element. In addition, in the literature on price discounts most scholars analyze from the perspective of buyers and suppliers, but they are often based on the basic assumption that product demand is a certain constant. The assumption that market demand information is asymmetric for each member of the supply chain is more in line with the actual situation. This paper will stand at the supplier's point of view and consider the assumption that market demand information is asymmetric for each member of the supply chain.

The closest papers to our work within the literature are Yoshida et al., 2014 and Ma et al., 2019 [13,25]. The former analyzes quantity discounts for multi-period production planning for supplier and retailer under demand uncertainty; however, it focuses on multi-stage problems and considers the monopoly environment. The latter study considers a competitive environment, but it builds a multi-channel supply chain. In this paper, we build a supply chain consisting of two competing suppliers and a retailer, and investigate the sustainable supplier's first-best price discount strategy and the competitive impact on the retailer.

## 3. Model Description

We consider a non-cooperative game, in which the supply chain consists of two suppliers ($m_i$, where $i = \{1, 2\}$) and one retailer ($r$). In order to simplify the problem, similar to Li and Wan (2017) we assume the retailer faces two ex-ante identical suppliers which provide homogeneous sustainable products with the same production cost, which is ignored in the calculation. Since the suppliers and the retailer are independent economic entities, they will determine their selling or purchasing strategies based on maximizing their own benefits or minimizing costs. According to the classic EOQ model, the retailer sends a total of $Q_0$ supply requirements to the two suppliers to supplement the inventory [16]. The ordering cost for each order is $S_r$, and the inventory cost of the unit product is $h_r$. Taking the supply chain stability into consideration, the retailer will order $Q_{0i}$ from the two suppliers separately, where $Q_{0i} > 0$ and $Q_{01} + Q_{02} = Q_0$. Before the price discount is provided, suppliers provide product at the wholesale price of $w_i$. The market demand rate is $D$, and its probability density function is $f(D)$, which is a random variable defined as $[L, M]$. Without considering price discount, the profit function of the supplier $m_i$ is $\pi_i = w_i q_i$, where $q_i$ denotes the quantity provided by $q_i$; the profit function of supplier $m_i$ under consideration of price discount is $\pi_i = (w_i - \sigma_i)q_i$, where $\sigma_i$ denotes the degree of wholesale price discount. Since the supplier does not directly contact the market, the supplier only knows the distribution of the market demand rate, while the retailer has full knowledge of the actual market demand rate. Without loss of generality, $m_1$ denotes the sustainable supplier, which offers a price discount to promote the sale of sustainable products, while $m_2$, as a competitive supplier, offers a price discount for competitive purposes. We assume that $m_2$ will not provide a price discount before $m_1$, which is also consistent with its competitive trait that $m_2$ will not actively offer price discounts unless $m_1$ threatens $m_2$ by offering a price discount.

A Stackelberg game is played between the suppliers and the retailer. The suppliers are the leader in the game relationship and the retailer is the follower. The game sequence is as follows: First, the suppliers decide whether to provide a price discount. Second, the suppliers decide the wholesale price and the quantity provided. Third, the retailer decides purchasing quantity according to the price and EOQ model. Finally, the demand is realized. Figure 1 describes the sequence of the events.

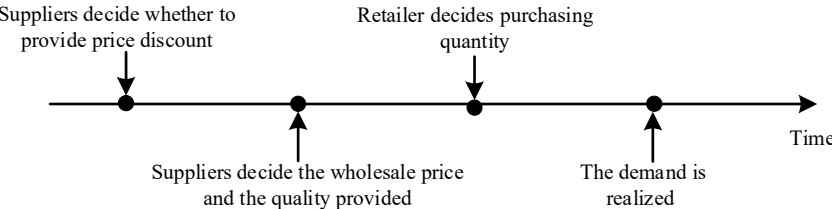

**Figure 1.** Sequence of Events.

We assume the price discount plan provided by the supplier $m_i$ is:

$$\begin{cases} w_i & Q_{0i} < \overline{Q}_i \\ w_i - \sigma_i & Q_{0i} \geq \overline{Q}_i \end{cases} \qquad (1)$$

This setting refers to Yoshida et al., 2014, where $Q_{0i}$ denotes the actual order quantity that is purchased by the retailer from the supplier $m_i$ ($i \in \{1, 2\}$), and $\overline{Q}_i$ is a constant, which denotes the critical value of the order quantity at which the retailer can enjoy a discounted price. In other words, when the retailer's order quantity $Q_{0i}$ at supplier $m_i$ is more than $\overline{Q}_i$, the supplier can get a discount ($\sigma_i$) on the original wholesale price ($w_i$), otherwise, they will buy at the original price ($w_i$). We assume a Cournot competition is played among suppliers, and the wholesale price of the sustainable supplier ($m_1$) and competitive supplier ($m_2$) are:

$$w_1 = D - q_1 - aq_2 \qquad (2)$$

$$w_2 = D - aq_1 - q_2 \tag{3}$$

This setting refers to Huang et al., 2018. $a$ denotes the difference between the two suppliers, $a \in [0, 1]$; to simplify the notation, we assume $a = 1$.

The scientific approach used in this paper can be concluded as follows: From the perspective of game theory, the Stackelberg game is played between suppliers and retailers and the Cournot game is played between the two suppliers; from the perspective of supply chain operation, price discount model, profit function model, and integral function are used. For ease of presentation, we summarize all of the notations in Table 1.

**Table 1.** Notations.

| Notations | Description |
|---|---|
| $m_i$ $Q_0$ | The supplier, where $i = 1$ when the supplier is the sustainable supplier and $i = 2$ when the supplier is the competitive supplier The total amount that the retailer sends to the two suppliers |
| $S_r$ | The ordering cost for each order |
| $h_r$ | The inventory cost of the unit product |
| $w_i$ | The wholesale price from $m_i$ before $m_i$ provides a price discount |
| $q_i$ $D$ | The quantity of product provided by $m_i$ The market demand rate |
| $\sigma_i$ | The degree of wholesale price discount of $m_i$ |
| $\pi_i^{NN}, \pi_i^{SN}, \pi_i^{SS}$ | The supplier $m_i$'s optimal profit, where superscript $NN$ denotes none of the suppliers provide a price discount; $SN$ denotes that only $m_1$ provides a price discount; $SS$ denotes both of the suppliers provide a price discount |
| $Q_{0i}$ | The actual order quantity that is purchased by the retailer from the supplier $m_i$ |
| $\overline{Q}_i$ | The critical value of the order quantity at which the retailer can enjoy a discounted price |
| $a$ | The difference between the two suppliers |

## 4. Analysis

We mainly discuss the optimal decision of the sustainable supplier ($m_1$) and the impact on the retailer. Following the backward induction, we start by analyzing the equilibrium of supply quantity and then move on to investigate the equilibrium price discount strategy.

### 4.1. The Equilibrium Quantity Strategy

In view of promoting sustainable product sales, $m_1$ considers providing the retailer with a price discount plan, and, considering that the sustainable supplier may offer a price discount, $m_2$ also has the motivation to offer a price discount. The discount competition between suppliers can be divided into three situations:

In the first situation, the two suppliers have no motivation to offer price discounts to the retailer. At this time, in order to increase the variety of commodities, the retailer proposes $\frac{Q_0}{2}$ to the two suppliers, respectively. The two suppliers determine the actual equilibrium supply according to the purchaser's order.

In the second situation, only the sustainable supplier ($m_1$) offers the price discount, while $m_2$ does not.

In the third situation, due to $m_1$ providing a price discount, the market share may be invaded. Because $m_2$ also has the intention to provide a price discount, there are two suppliers providing a price discount.

Situation 1: Neither supplier provides price discount ($NN$)

If the two suppliers do not provide price discounts, the retailer proposes $\frac{Q_0}{2}$ ($\sqrt{\frac{DS_r}{2h}}$) supply requirements to the two suppliers. Therefore, the payoff functions of the two suppliers at this time are:

$$\left\{ \begin{array}{l} \pi_1^{NN}(q_1) = (D - q_1 - q_2)q_1 \\ \pi_2^{NN}(q_2) = (D - q_1 - q_2)q_2 \end{array} \right. \tag{4}$$

In this situation, neither supplier provides a price discount, therefore, their wholesale price equals $D - q_1 - q_2$. Lemma 1 shows the equilibrium of suppliers' supply quantities under the situation in which neither supplier provides price discount.

**Lemma 1.** *When neither supplier provides price discount, the two suppliers' equilibrium quantity is* $(q_1^{NN}, q_2^{NN})$*, where* $(q_1^{NN}, q_2^{NN})$ *can be expressed as* $\left\{ \begin{array}{ll} \left(\frac{D}{3}, \frac{D}{3}\right) & L \leq D \leq \frac{9S_r}{2h} \\ \left(\sqrt{\frac{DS_r}{2h}}, \sqrt{\frac{DS_r}{2h}}\right) & \frac{9S_r}{2h} < D \leq M \end{array} \right.$.

**Proof.** According to the classical Cournot duopoly model, the two suppliers' equilibrium quantity strategy is $\left(\frac{D}{3}, \frac{D}{3}\right)$. We consider a non-cooperative game, the retailer will place an order, which equals $\sqrt{\frac{DS_r}{2h}}$, to each supplier according to the EOQ model, which means the final quantity that the suppliers can sell is $min\left\{\frac{D}{3}, \sqrt{\frac{DS_r}{2h}}\right\}$. Comparing $\frac{D}{3}$ and $\sqrt{\frac{DS_r}{2h}}$, we can get the two suppliers first-best supply quantity. □

Situation 2: Only one of the suppliers offers price discount (*SN*)

In this situation, $m_1$ provides the price discount while $m_2$ does not. This leads to two possible cases: (1) $Q_{01} < \overline{Q}_1$; (2) $Q_{01} \geq \overline{Q}_1$.

When $Q_{01} < \overline{Q}_1$, the retailer's order quantity is less than the discounted critical batch value, and the retailer will not be able to enjoy the price discount. Lemma 2 shows the equilibrium of suppliers' supply quantities in this case.

**Lemma 2.** *When only one supplier provides price discount* $(m_1)$*, if* $Q_{01} < \overline{Q}_1$*, the two suppliers' equilibrium quantity is* $(q_1^{SN}, q_2^{SN})$*, where* $(q_1^{SN}, q_2^{SN})$ *can be expressed as* $\left( D - \sigma_1 - \sqrt{\frac{2DS_r}{h}}, \right.$ $\left. 2\sqrt{\frac{2DS_r}{h}} - D + \sigma_1 \right)$*, and* $D \in \left( 0, \frac{4h\sigma_1 + 9S_r - 3\sqrt{9S_r^2 + 8\sigma_1 hS_r}}{4h} \right) \cup \left( \frac{4h\sigma_1 + 9S_r + 3\sqrt{9S_r^2 + 8\sigma_1 hS_r}}{4h}, +\infty \right)$.

**Proof.** When the retailer's economic order quantity is certain, due to supplier $m_1$ providing a price discount, the retailer will give priority to supplier $m_1$ for supply, and meet $Q_{02} = Q_0 - Q_{01}$. Therefore, according to the Cournot competition model, the equilibrium outcome can be obtained as $\left( D - \sigma_1 - \sqrt{\frac{2DS_r}{h}}, 2\sqrt{\frac{2DS_r}{h}} - D + \sigma_1 \right)$. Due to only $m_1$ offering a price discount, there must be $Q_{02} \leq Q_{01}$, from which the range of market demand $D$ can be obtained. □

Similarly, when $Q_{01} \geq \overline{Q}_1$, the order quantity of the retailer has exceeded the price discount threshold and the retailer can enjoy the price discount. Furthermore, in order to ensure order stability, the retailer must ensure that a certain number of products are purchased from supplier $m_2$, that is, $Q_{02} \geq 0$. Therefore, the payoff functions of the two suppliers at this time can be expressed as:

$$\left\{ \begin{array}{l} \pi_1^{SN}(\sigma, q_1) = (D - \sigma_1 - q_1 - q_2)q_1 \\ \pi_2^{SN}(q_2) = (D - q_1 - q_2)q_2 \end{array} \right. \tag{5}$$

In this situation, only $m_1$ provides the price discount while $m_2$ does not, therefore, the supplier $m_1$'s wholesale price equals $D - \sigma_1 - q_1 - q_2$, while the supplier $m_2$'s wholesale price equals $D - q_1 - q_2$, where $\sigma_1$ denotes the price discount provided by $m_1$. Lemma 3 shows the equilibrium of the suppliers' supply quantities in this case.

**Lemma 3.** *When only one supplier provides price discount ($m_1$), if $Q_{01} \geq \overline{Q}_1$, the two suppliers' equilibrium quantity is $\left(q_1^{SN}, q_2^{SN}\right)$, where $\left(q_1^{SN}, q_2^{SN}\right)$ can be expressed as*

$$\begin{cases} \left(\frac{D-2\sigma_1}{3}, \frac{D+\sigma_1}{3}\right) & \frac{2h\sigma_1 + 9S_r - 3\sqrt{9S_r^2 + 4\sigma_1 hS_r}}{4h} \leq D \leq \frac{2h\sigma_1 + 9S_r + 3\sqrt{9S_r^2 + 4\sigma_1 hS_r}}{4h} \\ \left(\sqrt{\frac{DS_r}{2h}}, \sqrt{\frac{DS_r}{2h}}\right) & \frac{2h\sigma_1 + 9S_r + 3\sqrt{9S_r^2 + 4\sigma_1 hS_r}}{4h} < D \leq M \end{cases}.$$

**Proof.** The retailer orders product according to the EOQ model. When $\sqrt{\frac{2DS_r}{h}} \geq q_1 + q_2$, that is, when the purchaser's order quantity exceeds the first-best supply capacity of the two suppliers, at this time, according to the classic Cournot game model, the supply quantity of the two suppliers can be calculated as $\frac{D-2\sigma_1}{3}$ and $\frac{D+\sigma_1}{3}$, the range of market demand $D$ can be obtained at the same time. On the contrary, when $\sqrt{\frac{2DS_r}{h}} < q_1 + q_2$, that is, when the supply of two suppliers is greater than the demand of the purchaser, the two suppliers can not provide $\frac{D-2\sigma_1}{3}$ and $\frac{D+\sigma_1}{3}$, respectively, so the retailer will still purchase $\sqrt{\frac{DS_r}{2h}}$ from the different suppliers separately. $\square$

Situation 3: Both suppliers offer price discount ($SS$)

In this case, both suppliers provide price discounts to the retailer. According to the principle of priority, we assume the retailer will give priority to the sustainable supplier $m_1$, then the payoff functions of the two suppliers at this time can be expressed as:

$$\begin{cases} \pi_1^{SS}(\sigma_1, q_1) = (D - \sigma_1 - q_1 - q_2)q_1 \\ \pi_2^{SS}(\sigma_2, q_2) = (D - \sigma_2 - q_1 - q_2)q_2 \end{cases} \tag{6}$$

In this situation, both $m_1$ and $m_2$ provide the price discount. Therefore, the supplier $m_1$'s wholesale price equals $D - \sigma_1 - q_1 - q_2$, and the supplier $m_2$'s wholesale price equals $D - \sigma_2 - q_1 - q_2$, where $\sigma_2$ denotes the price discount provided by $m_2$. Lemma 4 shows the equilibrium of the suppliers' supply quantities in this case.

**Lemma 4.** *When both suppliers provide price discount, the two suppliers' equilibrium quantity is $\left(q_1^{SS}, q_2^{SS}\right)$, where $\left(q_1^{SS}, q_2^{SS}\right)$ can be expressed as*

$$\begin{cases} \left(\frac{D+\sigma_2-2\sigma_1}{3}, \frac{D+\sigma_1-2\sigma_2}{3}\right) & H \leq D \leq G \\ \left(D - \sigma_1 - \sqrt{\frac{2DS_r}{h}}, 2\sqrt{\frac{2DS_r}{h}} + \sigma_1 - D\right) & G < D \leq M \end{cases},$$

*where $H = \frac{2[2h(\sigma_1+\sigma_2)+9S_r]-6\sqrt{9S_r^2+4hS_r(\sigma_1+\sigma_2)}}{8h}$, $G = \frac{2[2h(\sigma_1+\sigma_2)+9S_r]+6\sqrt{9S_r^2+4hS_r(\sigma_1+\sigma_2)}}{8h}$.*

**Proof.** The proof of Lemma 4 is similar to Lemma 3. $\square$

*4.2. The Equilibrium Price Discount Strategy*

In this subsection, we are going to investigate the sustainable supplier's first-best price discount strategy; in other words, determine the value of $\sigma_1$.

According to Lemma 2, when the retailer's original order quantity in the second stage is less than the discount threshold ($Q_{01} < \overline{Q}_1$), the expected profit of supplier $m_1$ can be denoted as:

$$J_{1.1}^{SN} = \int_L^{\frac{4h\sigma_1 + 9S_r - 3\sqrt{9S_r^2 + 8\sigma_1 hS_r}}{4h}} \left(x - \sqrt{\frac{2xS_r}{h}} - \sigma_1\right)^2 f(x)dx + \int_{\frac{4h\sigma_1 + 9S_r + 3\sqrt{9S_r^2 + 8\sigma_1 hS_r}}{4h}}^M \left(x - \sqrt{\frac{2xS_r}{h}} - \sigma_1\right)^2 f(x)dx \quad (7)$$

According to Lemma 3, when the retailer's original order quantity in the second stage is more than the discount threshold ($Q_{01} \geq \overline{Q}_1$), the expected profit of supplier $m_1$ can be denoted as:

$$J_{1.2}^{SN} = \int_{\frac{2h\sigma_1 + 9S_r - 3\sqrt{9S_r^2 + 4\sigma_1 hS_r}}{4h}}^{\frac{2h\sigma_1 + 9S_r + 3\sqrt{9S_r^2 + 4\sigma_1 hS_r}}{4h}} \frac{(x - 2\sigma_1)^2}{9} f(x)dx + \int_{\frac{2h\sigma_1 + 9S_r + 3\sqrt{9S_r^2 + 4\sigma_1 hS_r}}{4h}}^M \left(x - \sigma_1 - \sqrt{\frac{2xS_r}{h}}\right)\sqrt{\frac{xS_r}{2h}} f(x)dx \quad (8)$$

Similarly, according to Lemma 4, when the two suppliers provide price discounts at the same time, the expected profit of supplier $m_1$ can be denoted as:

$$J_1^{SS} = \int_{\frac{2h(\sigma_1+\sigma_2)+9S_r - 3\sqrt{9S_r^2 + 4(\sigma_1+\sigma_2)hS_r}}{4h}}^{\frac{2h(\sigma_1+\sigma_2)+9S_r + 3\sqrt{9S_r^2 + 4(\sigma_1+\sigma_2)hS_r}}{4h}} \frac{(x - 2\sigma_1 + \sigma_2)^2}{9} f(x)dx + \int_{\frac{2h(\sigma_1+\sigma_2)+9S_r + 3\sqrt{9S_r^2 + 4(\sigma_1+\sigma_2)hS_r}}{4h}}^M \left(x - \sigma_1 - \sqrt{\frac{2xS_r}{h}}\right)^2 f(x)dx \quad (9)$$

Lemma 5 summarizes the sustainable supplier's first-best price discount strategy.

**Lemma 5.** *For a given ordering cost $S_r$ and the inventory cost $h_r$ of the unit product. When only the sustainable supplier provides a price discount:*

*1. When the retailer's original order quantity in the second stage is less than the discount threshold, the sustainable supplier's first-best price discount is $\sigma_{1.1}^*$, where*

$$\sigma_{1.1}^* \triangleq \left\{ \sigma_{1.1}^{SN} : \left( \frac{9S_r - 3\sqrt{9S_r^2 + 8\sigma_1 hS_r}}{4h} + \sqrt{\frac{S_r\left(4h\sigma_1 + 9S_r - 3\sqrt{9S_r^2 + 8\sigma_1 hS_r}\right)}{2h^2}} \right)(D - \sigma_1 - \right.$$

$$\sqrt{\frac{S_r\left(4h\sigma_1 + 9S_r - 3\sqrt{9S_r^2 + 8\sigma_1 hS_r}\right)}{2h^2}}\right) f\left(\frac{9S_r - 3\sqrt{9S_r^2 + 8\sigma_1 hS_r}}{4h}\right)\left(1 - \frac{3S_r}{\sqrt{9S_r^2 + 8\sigma_1 hS_r}}\right) +$$

$$\int_L^{\frac{4h\sigma_1 + 9S_r - 3\sqrt{9S_r^2 + 8\sigma_1 hS_r}}{4h}} (2\sigma_1 - D - x)f(x)dx + \int_{\frac{2h\sigma_1 + 9S_r + 3\sqrt{9S_r^2 + 4\sigma_1 hS_r}}{4h}}^M (2\sigma_1 - D - x)f(x)dx =$$

$$\left( \frac{9S_r + 3\sqrt{9S_r^2 + 8\sigma_1 hS_r}}{4h} + \sqrt{\frac{S_r\left(4h\sigma_1 + 9S_r + 3\sqrt{9S_r^2 + 8\sigma_1 hS_r}\right)}{2h^2}} \right)(D - \sigma_1 -$$

$$\left. \sqrt{\frac{S_r\left(4h\sigma_1 + 9S_r + 3\sqrt{9S_r^2 + 8\sigma_1 hS_r}\right)}{2h^2}}\right) f\left(\frac{4\sigma_1 h + 9S_r + 3\sqrt{9S_r^2 + 8\sigma_1 hS_r}}{4h}\right)\left(1 + \frac{3S_r}{\sqrt{9S_r^2 + 8\sigma_1 hS_r}}\right) \right\}$$

*2. When the retailer's original order quantity in the second stage is more than the discount threshold, the sustainable supplier's first-best price discount is $\sigma_{1.2}^*$, where*

$$\sigma_{1.2}^* \triangleq \left\{ \sigma_{1.2}^{SN} : f\left(\frac{2h\sigma_1 + 9S_r + 3\sqrt{9S_r^2 + 4\sigma_1 hS_r}}{4h}\right)\left[ \frac{2h\sigma_1 + 9S_r + 3\sqrt{9S_r^2 + 4\sigma_1 hS_r}}{4h}(S_r - \right.\right.$$

$$\sqrt{\frac{S_r\left(2h\sigma_1 + 9S_r + 3\sqrt{9S_r^2 + 4\sigma_1 hS_r}\right)}{8}}\right)\left(1 + \frac{3S_r}{2\sqrt{9S_r^2 + 4\sigma_1 hS_r}}\right) - \sqrt{\frac{S_r\left(2h\sigma_1 + 9S_r + 3\sqrt{9S_r^2 + 4\sigma_1 hS_r}\right)}{8}}\left(\frac{1}{2} + \frac{3S_r}{2\sqrt{9S_r^2 + 4\sigma_1 hS_r}}\right)\frac{\sigma_1}{h}\right]$$

$$+ \int_{\frac{2h\sigma_1 + 9S_r + 3\sqrt{9S_r^2 + 4\sigma_1 hS_r}}{4h}}^M \sqrt{\frac{xS_r}{2h}} f(x)dx + \int_{\frac{2h\sigma_1 + 9S_r - 3\sqrt{9S_r^2 + 4\sigma_1 hS_r}}{4h}}^{\frac{2h\sigma_1 + 9S_r + 3\sqrt{9S_r^2 + 4\sigma_1 hS_r}}{4h}} \frac{8\sigma_1 - 4x}{9} f(x)dx =$$

$$\left( \frac{1}{2} + \frac{3S_r}{2\sqrt{9S_r^2 + 4\sigma_1 hS_r}} \right)\left[ \left( \frac{9S_r - 6h\sigma_1 - 3\sqrt{9S_r^2 + 4\sigma_1 hS_r}}{12h} \right)^2 f\left(\frac{2h\sigma_1 + 9S_r - 3\sqrt{9S_r^2 + 4\sigma_1 hS_r}}{4h}\right) - \right.$$

$$\left. \left( \frac{9S_r - 6h\sigma_1 + 3\sqrt{9S_r^2 + 4\sigma_1 hS_r}}{12h} \right)^2 f\left(\frac{2h\sigma_1 + 9S_r + 3\sqrt{9S_r^2 + 4\sigma_1 hS_r}}{4h}\right) \right]$$

*When the two suppliers provide a price discount at the same time, the sustainable supplier's first-best price discount is $\sigma_1^*$, where*

$$\sigma_1^* \triangleq \left\{ \sigma_1^{SS} : f\left( \frac{2h(\sigma_1+\sigma_2)+9S_r+3\sqrt{9S_r^2+4(\sigma_1+\sigma_2)hS_r}}{4h} \right)\left( \frac{1}{2}+ \right. \right.$$

$$\left. \frac{3S_r}{2\sqrt{9S_r^2+4(\sigma_1+\sigma_2)hS_r}} \right)\left[ \left( \frac{2h(\sigma_2-\sigma_1)+3S_r+\sqrt{9S_r^2+4(\sigma_1+\sigma_2)hS_r}}{4h} \right)^2 - \left( \frac{2h(\sigma_2-\sigma_1)+9S_r+3\sqrt{9S_r^2+4(\sigma_1+\sigma_2)hS_r}}{4h} + \right. \right.$$

$$\left. \left. \sqrt{\frac{2h(\sigma_2+\sigma_1)+9S_r+3\sqrt{9S_r^2+4(\sigma_1+\sigma_2)hS_r}}{2h^2}} \right)^2 \right] =$$

$$\left( \frac{6h(\sigma_2-\sigma_1)+9S_r-3\sqrt{9S_r^2+4(\sigma_1+\sigma_2)hS_r}}{2h} \right)^2 f\left( \frac{2h(\sigma_2+\sigma_1)+9S_r-3\sqrt{9S_r^2+4(\sigma_1+\sigma_2)hS_r}}{4h} \right)\left( \frac{1}{2} - \frac{3S_r}{2\sqrt{9S_r^2+4(\sigma_1+\sigma_2)hS_r}} \right) +$$

$$\frac{4}{9}\int_{\frac{2h(\sigma_1+\sigma_2)+9S_r-3\sqrt{9S_r^2+4(\sigma_1+\sigma_2)hS_r}}{4h}}^{\frac{2h(\sigma_1+\sigma_2)+9S_r+3\sqrt{9S_r^2+4(\sigma_1+\sigma_2)hS_r}}{4h}} (x-2\sigma_1+\sigma_2)f(x)dx+$$

$$\left. 2\int_{\frac{2h(\sigma_1+\sigma_2)+9S_r+3\sqrt{9S_r^2+4(\sigma_1+\sigma_2)hS_r}}{4h}}^{M} \left( x-\sigma_1-\sqrt{\frac{2xS_r}{h}} \right)f(x)dx \right\}$$

**Proof.** According to Formula (7), derivation of $J_{1.1}^{SN}$ with respect to $\sigma_1$, then we can get

$$\frac{\partial J_{1.1}^{SN}}{\partial \sigma_1} = \left( \frac{9S_r-3\sqrt{9S_r^2+8\sigma_1 hS_r}}{4h} + \sqrt{\frac{S_r\left(4h\sigma_1+9S_r-3\sqrt{9S_r^2+8\sigma_1 hS_r}\right)}{2h^2}} \right)(D-\sigma_1-$$

$$\sqrt{\frac{S_r\left(4h\sigma_1+9S_r-3\sqrt{9S_r^2+8\sigma_1 hS_r}\right)}{2h^2}} \right)f\left( \frac{4h\sigma_1+9S_r-3\sqrt{9S_r^2+8\sigma_1 hS_r}}{4h} \right)\left( 1-\frac{3S_r}{\sqrt{9S_r^2+8\sigma_1 hS_r}} \right)+$$

$$\int_L^{\frac{4h\sigma_1+9S_r-3\sqrt{9S_r^2+8\sigma_1 hS_r}}{4h}} (2\sigma_1-D-x)f(x)dx + \int_{\frac{2h\sigma+9S_r+3\sqrt{9S_r^2+4\sigma_1 hS_r}}{4h}}^{M} (2\sigma_1-D- \tag{10}$$

$$x)f(x)dx-\left( \frac{9S_r+3\sqrt{9S_r^2+8\sigma_1 hS_r}}{4h} + \sqrt{\frac{S_r\left(4h\sigma_1+9S_r+3\sqrt{9S_r^2+8\sigma_1 hS_r}\right)}{2h^2}} \right)(D-\sigma_1-$$

$$\sqrt{\frac{S_r\left(4h\sigma_1+9S_r+3\sqrt{9S_r^2+8\sigma_1 hS_r}\right)}{2h^2}} \right)f\left( \frac{4\sigma_1 h+9S_r+3\sqrt{9S_r^2+8\sigma_1 hS_r}}{4h} \right)\left( 1+\frac{3S_r}{\sqrt{9S_r^2+8\sigma_1 hS_r}} \right)$$

□

Then let Equation (10) equal zero, and the optimal price discount amount can be obtained as $\sigma_{1.1}^*$. Similarly, the optimal price discount can be obtained by derivation according to Equations (8) and (9), denoted as $\sigma_{1.2}^*$ and $\sigma_1^*$, respectively.

*4.3. The Impact on the Retailer*

Assume that the market size $D$ is uniformly distributed in $[0, 100]$. In view of the complexity of the expression, suppose $\sigma_1 = \sigma_2$ when both suppliers provide price discounts. This assumption conforms to the game theory: the two suppliers play a game based on the appropriate amount of discount, and the final result of the game is that the two suppliers provide same price discount.

**Lemma 6.** *The retailer's expected profit will be greater if the suppliers provide higher price discounts.*

**Proof.** According to the aforementioned assumption, the total cost of the retailer's order in the three situations can be denoted as follows:
1. Neither supplier provides price discount: The total cost of the retailer's order $c_1 = \frac{1}{100}\left( \int_L^{\frac{9S_r}{2h}} \frac{2xw}{3}f(x)dx + \int_{\frac{9S_r}{2h}}^{M} 2w\sqrt{\frac{xS_r}{2h}}f(x)dx \right);$
2. Only the sustainable supplier offers price discount: The total cost of the retailer's order

$$c_{2.1} = \frac{1}{100}\left( \int_0^{\frac{4h\sigma_1+9S_r-3\sqrt{9S_r^2+8h\sigma_1 S_r}}{4h}} \left[ w\sqrt{\frac{2xS_r}{h}} - \sigma_1\left(x-\sigma_1-\sqrt{\frac{2xS_r}{h}}\right)\right]f(x)dx + \int_{\frac{4h\sigma_1+9S_r-3\sqrt{9S_r^2+8h\sigma_1 S_r}}{4h}}^M \left[ w\sqrt{\frac{2xS_r}{h}} - \sigma_1\left(x-\sigma_1-\sqrt{\frac{2xS_r}{h}}\right)\right]f(x)dx\right);$$

$$c_{2.2} = \frac{1}{100}\left( \int_{\frac{2h\sigma_1+9S_r-3\sqrt{9S_r^2+4h\sigma_1 S_r}}{4h}}^{\frac{2h\sigma_1+9S_r+3\sqrt{9S_r^2+4h\sigma_1 S_r}}{4h}} \left[ \frac{w(2x-\sigma_1)}{3} - \frac{\sigma(x-2\sigma_1)}{3}\right]f(x)dx + \int_{\frac{2h\sigma_1+9S_r+3\sqrt{9S_r^2+4h\sigma_1 S_r}}{4h}}^M \sqrt{\frac{xS_r}{2h}}(2w-\sigma_1)f(x)dx\right);$$

3.     Both suppliers offer a price discount: The total cost of the retailer's order

$$c_3 = \frac{1}{100}\left( \int_{\frac{4h\sigma_1+9S_r-3\sqrt{9S_r^2+8h\sigma_1 S_r}}{4h}}^{\frac{4h\sigma_1+9S_r+3\sqrt{9S_r^2+8h\sigma_1 S_r}}{4h}} \left[ \frac{2(x-\sigma_1)(w-\sigma_1)}{3}\right]f(x)dx + \int_{\frac{4h\sigma_1+9S_r+3\sqrt{9S_r^2+8h\sigma_1 S_r}}{4h}}^M \sqrt{\frac{2xS_r}{h}}(w-\sigma)f(x)dx\right).$$

□

It is easy to find that $c_1 \geq c_{2.2} \geq c_3$, according to the retailer's profit function, and we can conclude that the expected benefit will be higher if the expected cost is lower.

## 5. Numerical Study

Let $h = 1500$; $S_r = 500$; $w = 3000$; $\sigma_1 \in [0, 1200]$. The corollary 1 describes the sustainable supplier's expected profit and the retailer's expected cost.

**Corollary 1.** *When only the sustainable supplier provides a price discount, the supplier's expected profit and the retailer's expected cost will increase with the addition of the price discount; when both suppliers provide a price discount, the supplier's expected profit and the retailer's expected cost will decrease with the price discount increase accordingly.*

**Proof.** Take all of the parameters into $J_{1.1}^{SN}$ and $J_{1.2}^{SN}$ respectively. $J_{1.1}^{SN} = \int_L^{\frac{4\sigma_1+3-\sqrt{9+24\sigma_1}}{4}}$ $\left(\sqrt{\frac{2x}{3}}+x-\sigma_1\right)\left(x-\sqrt{\frac{2x}{3}}-\sigma_1\right)f(x)dx + \int_{\frac{4\sigma_1+3+\sqrt{9+24\sigma_1}}{4}}^M \left(\sqrt{\frac{2x}{3}}+x-\sigma_1\right)\left(x-\sqrt{\frac{2x}{3}}-\sigma_1\right)$ $f(x)dx$; $J_{1.2}^{SN} = \int_{\frac{2\sigma_1+3-\sqrt{9+12\sigma_1}}{4}}^{\frac{4\sigma_1+3-\sqrt{9+24\sigma_1}}{4}} \frac{(x-2\sigma_1)^2}{9}f(x)dx + \int_{\frac{2\sigma_1+3+\sqrt{9+12\sigma_1}}{4}}^M \left(x-\sigma_1-\sqrt{\frac{2x}{3}}\right)\sqrt{\frac{x}{6}}f(x)dx$.

□

It is not difficult to see that $J_{1.1}^{SN}$ and $J_{1.2}^{SN}$ and increasing function of $\sigma_1$ by seeking the first derivative of $\sigma_1$.

Figure 2a,b reflect the effect of $\sigma_1$ on the sustainable supplier's expected profit when only one supplier provides price discounts and two suppliers both provide price discounts separately. In the case in which only the sustainable supplier offers price discounts, and another supplier has not taken any action, the greater the discount is provided, the more market share will be attracted. We take the supplier's 40% profit as the standard and set the maximum quantity discount to 1200 and, comparing with the Figure 2b, we find that the sustainable supplier's expected profit rise rate is greater when the original order quantity of the retailer is less than the critical value of the price discount provided by the supplier. When two suppliers both provide price discounts, according to the principle of game theory, there is a competitive relationship between the two suppliers. One party increases the discount and the other will follow closely. Then, the expected profit of the two suppliers will decline. On the one hand, both suppliers provide price discounts, and neither supplier will get more market share; on the other hand, as the quantity discounts continue to increase, the expected profit will also decline.

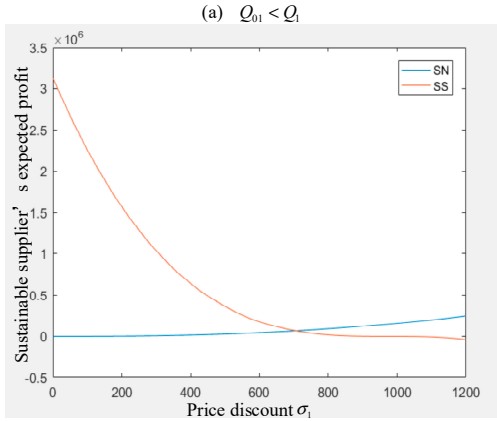
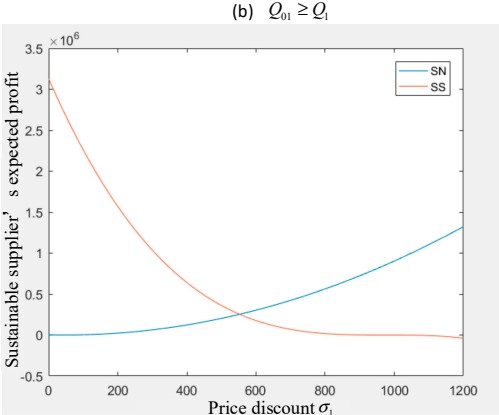

**Figure 2.** The impact of $\sigma_1$ on supplier's expected profit.

Figure 3a,b reflect the effect of $\sigma_1$ on the retailer's expected cost. According to Lemma 2, the quantity that the supplier $m_1$ provided will be less with the increase of $\sigma_1$; therefore, the retailer has to purchase more from the other supplier, and the retailer's expected cost will be higher. Comparing Lemma 2 and Lemma 3, it is not difficult to find that the influence of the change of $\sigma_1$ on the retailer is reduced when the retailer's original order quantity is greater than the price discount threshold provided by the supplier; therefore, the curve in Figure 3b is relatively flat. In addition, when two suppliers both provide price discounts, the two suppliers conduct a dynamic game with symmetric information. The price discounts will tend to be the same and continue to increase during a certain static period. This game process will also make the retailer's purchase cost continue to decrease.

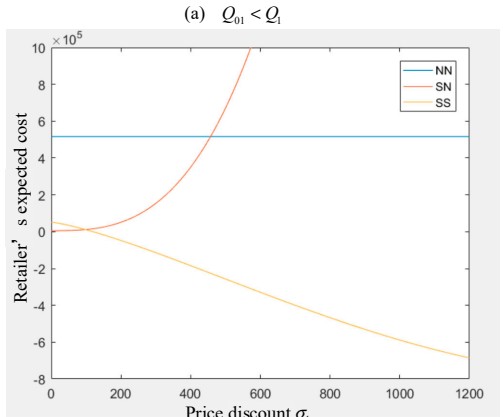
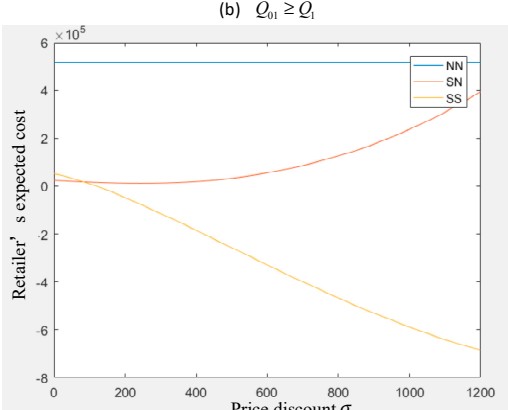

**Figure 3.** The impact of $\sigma_1$ on retailer's expected cost.

## 6. Conclusions

In order to promote sustainable product sales, many brands provide price discounts. In addition, price discounts are a common method for suppliers to compete for competitive advantage today. In response to this phenomenon, this paper establishes a supply chain model of two suppliers and a retailer: one of the suppliers aims to promote the demand of sustainable products by providing price discounts, and the other supplier is also motivated to provide price discounts to compete for market demand. Through building the Cournot model, we consider the balanced supply strategy and the optimal discount strategy from the perspective of the sustainable supplier, and analyze the effect of the discount on the supply chain members. We find that the sustainable supplier will be better off when they are the only supplier providing price discounts and will be worse off when both suppliers provide price discounts. Furthermore, when only the sustainable supplier provides price discounts, the retailer's purchasing cost will increase with the degree of the discount, while

the retailer's purchasing cost will decrease when both suppliers provide price discounts. In addition, taking the asymmetric information into consideration, this paper makes the research more reasonable and comprehensive by comparing the retailer's planned order volume and the supplier's critical value of price discount. In future research, we will promote this model into more suppliers corresponding to more retailers.

**Author Contributions:** Conceptualization, Y.L. and J.Z.; methodology, Y.L.; software, Y.L.; validation, Y.L. and J.Z.; formal analysis, Y.L.; investigation, Y.L.; resources, Y.L.; data curation, Y.L.; writing—original draft preparation, Y.L.; writing—review and editing, J.Z.; visualization, Y.L.; supervision, Y.L. and J.Z.; project administration, J.Z.; funding acquisition, J.Z. All authors have read and agreed to the published version of the manuscript.

**Funding:** This research was funded by the National Natural Science Foundation of China (71872036, 71832001), the Chinese Ministry of Education Project of Humanities and Social Sciences (18YJA630153), and the Fundamental Research Funds for the Central Universities (2232018H-07).

**Institutional Review Board Statement:** Not applicable.

**Informed Consent Statement:** Not applicable.

**Data Availability Statement:** The data used to support the findings of this study are included within the article.

**Acknowledgments:** Financial supports from the National Natural Science Foundation of China (71872036, 71832001), the Chinese Ministry of Education Project of Humanities and Social Sciences (18YJA630153), and the Fundamental Research Funds for the Central Universities (2232018H-07) are gratefully acknowledged.

**Conflicts of Interest:** The authors declare that they have no conflict of interest.

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
