# Peer review of "Sustainable Supplier’s Equilibrium Discount Strategy under Random Demand"

_sustainability, doi:10.3390/su14084802_

Round 1

Reviewer 1 Report

Thank you for the corrections

Reviewer 2 Report

It is an interesting topic. Well organized paper. The article is suitable for scientific publication.

This manuscript is a resubmission of an earlier submission. The following is a list of the peer review reports and author responses from that submission.

Round 1

Reviewer 1 Report

Good day and thank you for the opportunity to review your work. 

firstly I would like to state that there may be merit in the work but various aspects of the overall paper must be significantly improved. 

  1. the overall introduction is weak and does not explain or justify the study
  2. the literature review is very weak. 
    1. I don't find relevant literature to provide insights into the current body of knowledge
    2. I don't find literature on the status of supplier optimisation leading to identifying the knowledge gap of the work proposed
    3. literature is old and may have been superseded by new knowledge
  3. the reference list has not been checked and there are references cited but missing and vice versa. 
  4. the methodology section is not existent to a scientific approach and must be built and included
  5. there are various inconsistencies in the paper
    1. the actors quote environmentally friendly businesses at the beginning of section 3, but then based the model scenarios on none environmentally friendly suppliers.
    2. there are a few more that must be checked
  6. equations and maths. 
    1. here I find significant challenges with the layout and explanations as the flow is not articulated well.
    2. I find various inconsistencies starting with equation 1: theta is not defined, and the logic of Q denoting the critical value (should read quantity). I believe this may be the actual quantity. here the units are also mixed as w is the required amount and theta??
    3. there are various other challenges with interpreting the equations: use of convention is not explained and cannot be understood. 
    4. the equations, such as equation 4, is not mathematically correct, is PI a variable?
  7. I find the overall paper difficult to follow

Author Response

Responses to Reviewer 1’s Comments

Thank you for reviewing this manuscript and for the helpful comments. In this revision, we revised the paper according to your detailed comments. We believe that the revised version well addresses your concerns. For easy reference, all of the revisions are marked up by using the “Track Changes” function, such that any changes can be easily viewed.

Comment 1.   the overall introduction is weak and does not explain or justify the study

Response: Thanks for your suggestion. We have reorganized the research background and supplement the explanation of this paper in Chapter 1.

Comment 2.   the literature review is very weak.

  1. I don't find relevant literature to provide insights into the current body of knowledge
  2. I don't find literature on the status of supplier optimisation leading to identifying the knowledge gap of the work proposed
  3. literature is old and may have been superseded by new knowledge

Response: Thanks for your suggestion. This paper is related to the literature on the research of price discount and supply chain coordination. We have rearranged the above two parts of the literature, among which the most relevant literatures are Yoshida et al. 2014 and Ma et al. 2019. This paper refers to the mathematical models of these two papers, and places the issue of price discounts provided by suppliers in a competitive environment for discussion. In addition, we have added some new literature in related fields in the past two years.

Comment 3. the reference list has not been checked and there are references cited but missing and vice versa.

 Response: We have double checked the reference list and made necessary updates. Thanks for your suggestion.    

Comment 4. the methodology section is not existent to a scientific approach and must be built and included

Response: Thanks for your suggestion. This paper mainly uses the segmented pricing model and the supplier and retailer profit function model. We supplemented reference sources for key models to demonstrate scientific validity. This paper mainly refers to the mathematical models of Huang et al., 2018 and Yoshida et al. 2014, which are marked in the chapter 3.

Comment 5. there are various inconsistencies in the paper

  1. the actors quote environmentally friendly businesses at the beginning of section 3, but then based the model scenarios on none environmentally friendly suppliers.
  2. there are a few more that must be checked

Response: Thanks for your suggestion. This paper introduces the concept of “environment friendly” in Chapter 3 to explain the rationality of the existence of sustainable suppliers. Many brands in the market produce and sell sustainable products for the purpose of protecting the environment. In this paper, two suppliers are set to sell this sustainable product (both suppliers are environment friendly), but only one supplier offers price discounts in order to promote sustainable product sales, we define this supplier as “sustainable supplier”, and the other one is “competitive supplier”. We have made serious revisions to the full text to avoid this kind of misunderstanding.

Comment 6. equations and maths. 

    1. here I find significant challenges with the layout and explanations as the flow is not articulated well.
    2. I find various inconsistencies starting with equation 1: theta is not defined, and the logic of Q denoting the critical value (should read quantity). I believe this may be the actual quantity. here the units are also mixed as w is the required amount and theta??
    3. there are various other challenges with interpreting the equations: use of convention is not explained and cannot be understood. 
    4. the equations, such as equation 4, is not mathematically correct, is PI a variable?

Response: Thanks for your suggestion. First, this paper discusses the discount strategy of sustainable suppliers in three cases: 1. neither supplier offers price discounts; 2. only sustainable suppliers offer price discounts; 3. both suppliers offer price discounts. Then we further discuss the impact of suppliers' price discount strategy on supply chain members. Therefore, the layout of this paper is to first discuss the price discount strategies of sustainable suppliers in the three scenarios, and then discuss the impact of price discount strategy.

Secondly, the definition of parameters in this paper was not clear before, we have made corrections. Theta i represents the wholesale price discount given by the supplier mi. When the retailer's order quantity exceeds Qi, the price discount provided by the supplier mi can be enjoyed.

Finally, we supplement the definitions of some parameters in Chapter 3. Without the considering of price discount, the profit function of supplier  is ; the profit function of supplier  under the consideration of price discount is , where  denotes the degree of wholesale price discount.  denotes supplier ’s profit.

Comment 5. I find the overall paper difficult to follow

Response: Thanks for your suggestion. The original text has shortcomings in all aspects. According to your review comments, we have made serious revisions. We hope that the revised paper will be easier to understand.

Reviewer 2 Report

1) Writing language must be improved. All sentences must be passive.
2) Abstract must be improved.  It is too short and there is no specific information about the research.
3) Conclusion must be improved. 
4) A comparison part must be added to the research.
5) The following questions must be answered in the manuscript. Why did you choose this approach? What is the main difference between this approach and others? What is the main advantage of the proposed approach? What are the strong parts and weak parts of the proposed approach?
6) Originality of model must be mentioned.
7) Must be added references in parts 3 and 4.

Author Response

Responses to Reviewer 2’s Comments

Thank you for reviewing this manuscript and for the helpful comments. In this revision, we revised the paper according to your detailed comments. We believe that the revised version well addresses your concerns. For easy reference, all of the revisions are marked up by using the “Track Changes” function, such that any changes can be easily viewed.

Comment 1.   Writing language must be improved. All sentences must be passive.

Response: Thanks for your suggestion. We have carefully revised the English expression of this paper.

Comment 2.   Abstract must be improved.  It is too short and there is no specific information about the research.

Response: Thanks for your suggestion. We have revised the Abstract section by adding key research methods and important conclusions

Comment 3. Conclusion must be improved.

 Response: Thanks for your suggestion. We have carefully revised the Conclusion section, summarized the main research findings and supplemented future research direction.

Comment 4. A comparison part must be added to the research.

Response: This paper discusses the discount strategy of sustainable suppliers in three cases: 1. neither supplier offers price discounts; 2. only sustainable suppliers offer price discounts; 3. both suppliers offer price discounts. Then we further discuss the impact of suppliers' price discount strategy on supply chain members. In Chapter 5, we compare the sustainable supplier’s profit and retailer’s purchasing cost under different situation.

Comment 5. The following questions must be answered in the manuscript. Why did you choose this approach? What is the main difference between this approach and others? What is the main advantage of the proposed approach? What are the strong parts and weak parts of the proposed approach?

Response: Thanks for your suggestion. We supplement the explanation of above questions in the paper. We set the model of this paper based on the actual business model and the classic economic model. This paper mainly refers to the mathematical models of Huang et al., 2018 and Yoshida et al. 2014, which are marked in the chapter 3. The main difference of this paper is that we put the sustainable supplier’s price discount strategy under the competitive environment, the other papers discuss monopoly environments and dual-channel supply chains, respectively. This model setting is more in line with the supply chain model structure of this paper. However, there is a problem that the calculation is complex and an analytical solution cannot be obtained. Therefore, we draw relevant management conclusions through the numerical study in Chapter 5.

Comment 6. Originality of model must be mentioned.

Response: Thanks for your suggestion. We supplemented this part of the explanation in Chapter 2, comparing the differences between this paper and the main references, which can reflect the originality of this paper.

Comment 6. Must be added references in parts 3 and 4.

Response: Thanks for your suggestion. We add some references in Chapter 3 and 4.

Round 2

Reviewer 1 Report

Good day

i unfortunately dont find the corrections requested.

the paper needs substantive correction.

Author Response

Responses to Reviewer 1’s Comments

Thank you for reviewing this manuscript and for the helpful comments. In this revision, we revised the paper according to your detailed comments. We believe that the revised version well addresses your concerns. For easy reference, all of the revisions are marked up by using the “Track Changes” function, such that any changes can be easily viewed. Blue font indicates the modified part of the original text.

Comment: I unfortunately don’t find the corrections requested. The paper needs substantive correction.

Response: Thanks for your suggestion. The original text has shortcomings in all aspects. According to your review comments, we have made serious revisions. We are so sorry that the first round of revisions did not satisfy you. This time, we have made necessary revisions and explained your questions in the first round of revisions in detail.

Comment 1.   the overall introduction is weak and does not explain or justify the study

Response: Thanks for your suggestion. We have reorganized the research background and supplement the explanation of this paper in Chapter 1. For the consideration of sustainable development, many companies have launched sustainable products. In order to increase the sales of sustainable products, firms consider propose price discount strategies (we define them sustainable firms). On the other hand, sustainable firms’ price discounts may also bring competitive pressures on the other firms, therefore, other firms also have the motivation to provide price discount (we define them competitive firms). In Section 1, we first introduce the background that many sustainable firms provide price discount (the first paragraph in Section 1), then we introduce the fact that many firms provide price discount due to the competitive pressure (the second paragraph in Section 1). The third and fourth paragraphs are supplemented according to the reviewers' comments, introducing the research questions and research purposes of this paper.

The supplementary content is:

In this paper, we are going to discuss the impact of quantity discounts, which is adopted for the two different purposes, on sustainable supplier: one is to drive sustainable product sales, the other is for competitive purpose. The competitive environment makes sustainable supplier face a trade-off when she decides whether to provide price discount: on the one hand, when a price discount is provided, retailers may purchase more from the sustainable supplier; on the other hand, other suppliers may also provide price discount for competitive purpose when they are informed that the sustainable supplier will provide price discount: if they do, the competition of price discount will reduce each supplier’s sales profits finally, but it can avoid the market being divided up; if not, the opposite is true(McKechnie et al., 2012; Luo et al., 2014).

Based on the above considerations, we develop a model in which a sustainable supplier considers providing price discount to the retailer to promote the sustainable product sale. There has another supplier provide the same product and may also has the motivation to provide price discount when it gets informed that the sustainable supplier will provide price discount. We explore the sustainable supplier’s first-best quantity decision and price discount strategy, then we investigate the influence of suppliers’ price discount on those supply chain member. We find that when only sustainable supplier provides price discount, his expected profit will increase due to the retailer will order more from here, while the sustainable supplier’s expected profit will decrease when both suppliers provide price discount, because the game result of two suppliers on price discount is that both sides offer the same price discount, which leads to the sustainable supplier lowering the price on the one hand, and the market demand does not increase on the other hand, while the retailer will be better off in this competitive game.

Comment 2.   the literature review is very weak.

  1. I don't find relevant literature to provide insights into the current body of knowledge
  2. I don't find literature on the status of supplier optimisation leading to identifying the knowledge gap of the work proposed
  3. literature is old and may have been superseded by new knowledge

Response: Thanks for your suggestion.

  1. The two papers that most relevant to our work are Yoshida et al. 2014 and Ma et al. 2019. The former one analyzes quantity discounts for multi-period production planning for supplier and retailer under demand uncertainty. But it focuses on multi-stage problems, and it considers the monopoly environment. The later one considers a competitive environment, but it builds a multi-channel supply chain. In this paper, we build a supply chain consisting of two competing suppliers and a retailer, and investigate the sustainable supplier’s first-best price discount strategy and the competitive impact on the retailer. (The fourth paragraph in Section 2)

We supplement the references in mathematical models: for example, Formula 1 refers to Yoshida et al. 2014; Formula 2 and 3 refer to Huang et al., 2018. All of the references are supplemented in Section 3.

2.This paper is related to the literature on the research of price discount and supply chain coordination. We have rearranged the above two parts of the literature. The literature on the status of supplier optimization is not very related with our work, in fact, when solving the supply chain manage problem, most of the researches are aim to find the optimal strategy, but in this paper we more concentrate on the issue of coordination and focus on coordination between suppliers, therefore, we only present a literature review of two parts: price discounting and supply chain coordination.

  1. In addition, we have added some new literature in related fields in the past two years.

Comment 3. the reference list has not been checked and there are references cited but missing and vice versa.

 Response: We have double checked the reference list and made necessary updates. Thanks for your suggestion.    

Comment 4. the methodology section is not existent to a scientific approach and must be built and included

Response: Thanks for your suggestion. This paper mainly uses the segmented pricing model and the supplier and retailer profit function model. We supplemented reference sources for key models to demonstrate scientific validity. This paper mainly refers to the mathematical models of Huang et al., 2018 and Yoshida et al. 2014, which are marked in the chapter 3:

We assume the price discount plan provided by the supplier  is:

                           (1)

This setting refers to Yoshida et al. 2014.

We assume a Cournot competition is played among suppliers, the wholesale price of the sustainable supplier ( ) and competitive supplier ( ) are:

                           (2)

                           (3)

This setting refers to Huang et al., 2018.

Comment 5. there are various inconsistencies in the paper

  1. the actors quote environmentally friendly businesses at the beginning of section 3, but then based the model scenarios on none environmentally friendly suppliers.
  2. there are a few more that must be checked

Response: Thanks for your suggestion. We have revised the full text based on your comments. This paper introduces the concept of “environment friendly” in Chapter 3 to explain the rationality of the existence of sustainable suppliers. Many brands in the market produce and sell sustainable products for the purpose of protecting the environment. In this paper, two suppliers are set to sell this sustainable product (both suppliers are environment friendly), but only one supplier offers price discounts in order to promote sustainable product sales, we define this supplier as “sustainable supplier”, and the other one is “competitive supplier”. We have made serious revisions to the full text to avoid this kind of misunderstanding.

Comment 6. equations and maths. 

    1. here I find significant challenges with the layout and explanations as the flow is not articulated well.
    2. I find various inconsistencies starting with equation 1: theta is not defined, and the logic of Q denoting the critical value (should read quantity). I believe this may be the actual quantity. here the units are also mixed as w is the required amount and theta??
    3. there are various other challenges with interpreting the equations: use of convention is not explained and cannot be understood. 
    4. the equations, such as equation 4, is not mathematically correct, is PI a variable?

Response: Thanks for your suggestion. First, this paper discusses the discount strategy of sustainable suppliers in three cases: 1. neither supplier offers price discounts; 2. only sustainable suppliers offer price discounts; 3. both suppliers offer price discounts. Then we further discuss the impact of suppliers' price discount strategy on supply chain members. Therefore, the layout of this paper is to first discuss the price discount strategies of sustainable suppliers in the three scenarios, and then discuss the impact of price discount strategy.

Secondly, the definition of parameters in this paper was not clear before, we have made corrections. Theta i represents the wholesale price discount given by the supplier mi. When the retailer's order quantity exceeds Qi, the price discount provided by the supplier mi can be enjoyed.

Finally, we supplement the definitions of some parameters in Chapter 3. Without the considering of price discount, the profit function of supplier  is ; the profit function of supplier  under the consideration of price discount is , where  denotes the degree of wholesale price discount.  denotes supplier ’s profit.

In addition to the revisions of the previous round of review, this round continued to revise the relevant mathematical models in Chapter 4

Comment 7. I find the overall paper difficult to follow

Response: Thanks for your suggestion. The original text has shortcomings in all aspects. According to your review comments, we have made serious revisions. We hope that the revised paper will be easier to understand.

Reviewer 2 Report

The authors made the necessary corrections. The work may be published as it is.

Author Response

Thank you very much for your valuable comments and thank you very much for your recognition.